Seamens’ Sign: a novel electrocardiogram prediction tool for left ventricular hypertrophy

Walker Philip philip.w.walker@gmail.com 1
Jenkins Cathy A. 2
Hatcher Jeremy 3
Freeman Clifford 1
Srica Nickolas 1
Rosell Bryant 1
Hanna Eriny 1
March Cooper 3
Seamens Charles 1
Storrow Alan 1
McCoin Nicole 1
1 Department of Emergency Medicine, Vanderbilt University , Nashville , TN , United States of America
2 Department of Biostatistics, Vanderbilt University , Nashville , TN , United States of America
3 School of Medicine, Vanderbilt University , Nashville , TN , United States of America
Sergi Consolato
Electronic publication date: 2022 May 31
Publication date: 2022
Volume: 10
Electronic Location ID: e13548
Received 2022 Feb 28; Accepted 2022 May 16
Copyright: ©2022 Walker et al.
Copyright year: 2022
Copyright holder: Walker et al.
License: This is an open access article distributed under the terms of the Creative Commons Attribution License, which permits unrestricted use, distribution, reproduction and adaptation in any medium and for any purpose provided that it is properly attributed. For attribution, the original author(s), title, publication source (PeerJ) and either DOI or URL of the article must be cited.
License URL: https://creativecommons.org/licenses/by/4.0/

Keywords: Electrocardiogram, Left ventricular hypertrophy

Funding: The authors received no funding for this work.

==============================
Introduction

Patients with left ventricular hypertrophy (LVH) diagnosed by electrocardiogram (ECG) have increased mortality and higher risk for life-threatening cardiovascular disease. ECGs offer an opportunity to identify patients with increased risk for potential risk-modifying therapy. We developed a novel, quick, easy to use ECG screening criterion (Seamens’ Sign) for LVH. This new criterion was defined as the presence of QRS complexes touching or overlapping in two contiguous precordial leads.

Methods

This study was a retrospective chart review of 2,184 patient records of patients who had an ECG performed in the emergency department and a transthoracic echocardiogram performed within 90 days. The primary outcome was whether Seamens’ Sign was noninferior in confirming LVH compared to other common diagnostic criteria. Test characteristics were calculated for each of the LVH criteria. Inter-rater agreement was assessed on a random sample using Cohen’s Kappa.

Results

Median age was 63, 52% of patients were male and there was a 35% prevalence of LVH by transthoracic echocardiogram (TTE). Nine percent were positive for LVH on ECG based on Seamens’ Sign. Seamens’ Sign had a specificity of 0.92. Tests assessing noninferiority indicated Seamens’ Sign was non-inferior to all criteria (p < 0.001) except for Cornell criterion for women (p = 0.98). Seamens’ Sign had 90% (0.81–1.00) inter-rater agreement, the highest of all criteria in this study.

Conclusion

When compared to both the Sokolow-Lyon criteria and the Cornell criterion for men, Seamens’ Sign is noninferior in ruling in LVH on ECG. Additionally, Seamens’ Sign has higher inter-rater agreement compared to both Sokolow-Lyon criteria as well as the Cornell criteria for men and women, perhaps related to its ease of use.

Introduction

Patients with left ventricular hypertrophy (LVH) diagnosed by electrocardiogram (ECG) have increased mortality and a higher risk for life-threatening cardiovascular disease, most commonly coronary artery disease in men and heart failure in women (Sundström et al., 2001; Desai, Ning & Lloyd-Jones, 2012). LVH is commonly diagnosed via ECG in the emergency department (ED), often in cardiac workup or incidentally (Shoenberger et al., 2009). These LVH diagnoses provide evidence of patients’ overall cardiovascular health and inform cardiac risk management and stratification (Bacharova & Estes, 2017).

The process of diagnosing LVH is multi-faceted. Cardiac echocardiography or left ventricular mass measurements via cardiac magnetic resonance imaging (MRI) are the gold standards (Lang et al., 2015; Zhang et al., 2019). However, widespread LVH screening via these methods is neither feasible nor cost effective. ECGs performed in the ED setting offer an opportunity to identify patients with increased risk for cardiac mortality and who are candidates for potential risk-modifying therapy. Despite the importance of efficient, accurate LVH prediction based on ECGs, commonly used methods have known diagnostic test inaccuracies and are challenging to use (Hedman et al., 2020).

Most current criteria require measuring or adding varying lead voltages, and may be complicated by risk of calculation errors. There are numerous ECG criteria for identifying LVH, with varying tests characteristics influenced by underlying cardiac conduction defects, gender, race, and body habitus (Antikainen et al., 2006; Jaggy et al., 2000; Germano, 2015).

Ultimately, ECG diagnostic criteria for LVH are clinically lacking. Though many attempts at defining more sensitive and specific ECG criteria for LVH have been proposed, none approach the accuracy of gold-standard imaging modalities (Mahn et al., 2014). These proposed criteria generally increase complexity to marginally improve sensitivity and/or specificity, creating a barrier to quick application in the ED setting.

Instead of more complex ECG criteria, we propose a novel, quick, and easily recognizable screening criterion for LVH can be learned with little memorization and applied in a fast-paced setting. The proposed criterion—“Seamens’ Sign”—involves one question: do any of the QRS complexes in the precordial leads of a standard 12-lead ECG touch or cross another QRS complex (e.g., V1 QRS complex touching/crossing V2 QRS complex) (Fig. 1)?

Figure 1 Standard 12-lead electrocardiogram demonstrating Seamens’ Sign with precordial QRS complexes overlapping and/or touching.

This is best seen with V2 touching/overlapping V3, as well as V4 touching/overlapping V5.

From an electrophysiologic standpoint, many reasons explain why precordial lead QRS complex touching/overlap works for LVH detection. Typical, non-pathologic R wave progression in the precordial leads shows that as the electrical signal passes from the atrioventricular node towards the apex of the left ventricle, prominent S waves (overall negative deflection) in V1 and V2 transition to predominant R waves (overall positive deflection) in V5 and V6. As the left ventricle hypertrophies, changes occur leading to an electrical vector of greater magnitude, translating to increased amplitude of the S and R waves in the precordial leads, often leading to the precordial QRS complexes touching or overlapping (Hill, 2012; Bacharova et al., 2010).

In this study, we evaluated the test characteristics of the proposed Seamens’ Sign and compared its ability to confirm an LVH diagnosis against three of the most used voltage criteria today (two Sokolow-Lyon criteria and the Cornell Criteria).

Methods

Study design

This study was an electronic health record (EHR) retrospective chart review at a quaternary care academic medical center. The data collection period included clinical tests performed 5 July 2019 through 14 January 2020 as part of routine ED care. This study was reviewed by the Vanderbilt University institutional review board (IRB) and given an exemption from full IRB review and informed consent given its retrospective nature and no identifying protected health information was kept (IRB#200150).

Eligibility

A query of the EHR was performed, identifying consecutive patients with both an ECG and a transthoracic echocardiogram (TTE) performed within 90 days of each other. No patients were excluded prior to data analysis on the basis of age, ethnicity, comorbidities, co-existing cardiac diagnoses evident on ECG or TTE, or other clinical factors.

Data collection & ECG coding

Total sample size for chart review was determined based on the number of subjects needed to estimate the sensitivity of Seamens’ Sign to a pre-specified margin of error. A total of 2,184 patient records were reviewed based on estimating a hypothesized sensitivity of 65% to a 2% margin of error. Data gathered during the initial EHR query included age, sex, ECG time/date, and TTE time/date. Each patient chart was assembled by an initial set of reviewers (primarily third year medical students), and assigned a random study number. They downloaded a copy of the ECG labeled with the study number and with all patient identifiers removed. They reviewed the TTE report and recorded whether or not LVH was identified. LVH was defined as any mention of the patient having concentric LVH in the TTE report. Septal or other focal hypertrophy was not considered LVH.

A second, independent set of blinded reviewers (Emergency Medicine residents) reviewed each ECG for signs of LVH based on two Sokolow-Lyon criteria, the Cornell criteria, and the study criterion, Seamens’ Sign. The first criterion, noted as the Sokolow-Lyon 1 criterion (SL-1) was defined as the S wave in lead V1 plus the R wave in lead V5 or V6 (using larger R wave in V5 or V6) being greater than or equal to 35 mm. The second Sokolow-Lyon criterion, noted as Sokolow-Lyon 2 criterion (SL-2), was defined as the R wave in lead aVL being greater than or equal to 11 mm. The Cornell criteria were defined as the S wave in lead V3 plus the R wave in lead aVL being greater than 28 mm in males or greater than 20 mm in females. These criteria’s test characteristics were compared against the test characteristics for the proposed new criterion, Seamens’ Sign. This new criterion was defined as the presence of QRS complexes touching or overlapping in two contiguous precordial leads (lead V1 QRS complex touching/crossing lead V2 QRS complex, or lead V2 QRS complex touching/crossing lead V3 QRS complex, or lead V4 QRS complex touching/crossing lead V5 QRS complex, or lead V5 QRS complex touching/crossing lead V6 QRS complex).

A total of 250 patient records were randomly selected to be re-reviewed by the second set of blinded reviewers in order to evaluate inter-rater agreement. These patient records were distributed to the blinded reviewers to ensure that no patient record was reviewed by the same blinded reviewer twice. The blinded reviewers re-reviewed this subset of ECGs as previously described.

Outcome measures

The primary outcome was determining whether Seamens’ Sign was noninferior in confirming LVH compared to the other criteria.

Analysis

Diagnosis of concentric LVH by TTE was considered the gold standard against which the various ECG criteria for LVH were compared to determine sensitivity and specificity.

Descriptive statistics of demographic and clinical characteristics were computed for the study population. Test characteristics, including sensitivity, specificity, and positive and negative predictive values, along with their 95% confidence intervals, were calculated for each of the LVH criteria. Non-inferiority of Seamens’ Sign criterion compared to the Cornell and Sokolow-Lyon criteria was evaluated using a method specified in a 2002 manuscript published in Statistics in Medicine designed for paired binary data (Liu et al., 2002). The margin of non-inferiority was pre-specified at 5% (p = 0.05). To compare Cornell criteria for men and women to those with Seamens’ Sign, only men with Seamens’ Sign were compared to other men meeting Cornell criteria, and the same method was used for women. To ensure validity of reviewer ECG coding and assess ease of interpretation, inter-rater agreement was assessed on a random sample using Cohen’s Kappa with the 95% confidence intervals. All statistical analyses were performed using R statistical programming language, Version 3.5.2.

Results

Patient characteristics

Patient characteristics are listed in Table 1. Median age was 63, 52% of patients were male and there was a 35% prevalence of LVH by TTE. The vast majority of TTEs were performed within 1 day of ECGs, with the median of 1 day, and the interquartile range of 0 to 21 days. Nine percent were positive for LVH on ECG based on Seamens’ Sign, and 3% and 7% were positive for LVH on ECG based on Sokolow-Lyon 1 and Sokolow-Lyon 2 criteria, respectively. There were 7% of men and 13% of women positive for LVH on ECG based on Cornell criteria.

Table 1 Descriptive statistics on demographic and clinical characteristics of the cohort.

	N	N= 2,184	
Age	2,184	63 (51, 73)	
Sex	2,184		
Male		52% (1,135)	
Female		48% (1,049)	
TTE for LVH (gold standard)	2,184		
No		65% (1,428)	
Yes		35% (756)	
ECG to TTE (days)	2,184	1 (0, 21)	
Seamens’ Sign positive for LVH	2,184		
No		91% (1,994)	
Yes		9% (190)	
Sokolow-Lyon 1 positive for LVH	2,184		
No		97% (2,113)	
Yes		3% (71)	
Sokolow-Lyon 2 positive for LVH	2,184		
No		93% (2,037)	
Yes		7% (147)	
Cornell (overall) positive for LVH	2,184		
No		90% (1,971)	
Yes		10% (213)	
Cornell (men) positive for LVH	1,135		
No		93% (1056)	
Yes		7% (79)	
Cornell (women) positive for LVH	1,049		
No		87% (914)	
Yes		13% (135)	
Notes.

N is the number of non-missing values. Numbers after proportions are frequencies, with the exception of age and ECG to TTE. Age and ECG to TTE are reported as the median, with following numbers the lower and upper interquartile for these continuous variables.

Sensitivity and specificity

Test characteristics are presented in Table 2. Specificities ranged from 0.89 for the Cornell criterion for women to 0.98 for the Sokolow-Lyon 1 criterion, with Seamens’ Sign having a specificity of 0.92.

Table 2 Test characteristics for Seamens’ Sign criterion, Sokolow-Lyon 1 (SL-1) and 2 (SL-2) criteria, and Cornell criteria for diagnosing left ventricular hypertrophy from electrocardiograms.

Test	Sensitivity	Specificity	PPV	NPV	
	Estimate	95% CI	Estimate	95% CI	Estimate	95% CI	Estimate	95% CI	
Seamens’ Sign	0.11	(0.09, 0.13)	0.92	(0.91, 0.94)	0.43	(0.36, 0.51)	0.66	(0.64, 0.68)	
SL-1	0.05	(0.03, 0.07)	0.98	(0.97, 0.98)	0.51	(0.39, 0.63)	0.66	(0.64, 0.68)	
SL-2	0.08	(0.06, 0.10)	0.94	(0.93, 0.95)	0.41	(0.33, 0.50)	0.66	(0.64, 0.68)	
Cornell Overall	0.13	(0.11, 0.15)	0.92	(0.90, 0.93)	0.46	(0.39, 0.52)	0.67	(0.64, 0.69)	
Cornell Men	0.09	(0.07, 0.12)	0.94	(0.92, 0.96)	0.51	(0.39, 0.62)	0.62	(0.59, 0.65)	
Cornell Women	0.18	(0.14, 0.23)	0.89	(0.87, 0.91)	0.42	(0.34, 0.51)	0.71	(0.68, 0.74)	
Notes.

Abbreviations CI confidence interval

PPV positive predictive value

NPV negative predictive value

Non-inferiority

Tests assessing noninferiority indicated Seamens’ Sign was non-inferior to all criteria (p < 0.001) except for Cornell criterion for women (p = 0.98) (Table 3).

Table 3 p-values for tests assessing non-inferiority of Seamens’ Sign when compared to other commonly used tests.

Comparison	P	
Sokolow-Lyon 1	<0.001	
Sokolow-Lyon 2	<0.001	
Cornell Overall	<0.001	
Cornell Men	<0.001	
Cornell Women	0.98	

Inter-rater agreement

Inter-rater agreement was assessed on 250 subjects using Cohen’s Kappa statistic and a 95% confidence interval (Table 4). Seamens’ Sign had 90% (0.81–1.00) agreement, the highest of all criteria, attributed to its quick application and ease of use. Sokolow-Lyon 1 and Sokolow-Lyon 2 had inter-rater agreement of 65% (0.40–0.91) and 87% (0.75–1.00) respectively. Sokolow-Lyon 1 likely has lower inter-rater agreement secondary to multiple leads used and subjectivity in selecting the R wave in lead V5 or V6. Cornell criteria for men and women had inter-rater agreements of 76% (0.56–0.96) and 79% (0.62–0.97), respectively.

Table 4 Inter-rater agreement using Cohens Kappa with 95% confidence interval.

Test	Kappa	95% CI	
Seamens’ Sign	0.9	(0.81, 1.00)	
Sokolow-Lyon 1	0.65	(0.40, 0.91)	
Sokolow-Lyon 2	0.87	(0.75, 1.00)	
Cornell Overall	0.82	(0.69, 0.94)	
Cornell (Men)	0.76	(0.56, 0.96)	
Cornell (Women)	0.79	(0.62, 0.97)	

Discussion

While modalities other than ECG are the gold-standard for diagnosing LVH, it is important to account for their difficulty and cost compared to the quick, easy to obtain, and inexpensive ECG, particularly in emergency care settings. Furthermore, there are data suggesting LVH diagnosed by ECG criteria represents a clinically distinct entity, and has been associated with increased mortality and other pathologic conditions (Aro & Chugh, 2016). This furthers the argument of the importance of fast, reliable methods of diagnosing LVH by ECG.

This analysis suggests Seamens’ Sign is non-inferior to other methods of evaluating LVH on ECG, and has high inter-rater agreement. It is easy to perform quickly without a measurement device or need for any calculations at all. Given these findings, we believe that Seamens’ Sign is easily applicable in emergency care settings and can facilitate the diagnosis of LVH, potentially leading to decreased cardiac morbidity and mortality.

Strengths

This is a large study comparing test characteristics of multiple criteria to Seamens’ Sign.

Three of the most widely used criteria were chosen to model real-world application. Compared to most prior studies, the number of subjects analyzed was larger. Of the prior 14 studies analyzing ECG diagnosis of LVH, enrollment ranged from 94 to 5608 patients; this study is the third largest. Those interpreting ECGs were blinded from the TTE results to remove any bias. The proliferative phase of cardiac remodeling takes place within the first 2–7 days after a myocardial infarction, transitioning to the maturing phase around day 7 (French & Kramer, 2007). Based on these findings, a 90-day limit on the time difference between the TTE and ECG dates was placed to reduce the likelihood of cardiac remodeling affecting results. The majority of the ECGs and TTEs were performed within 1 day of each other, limiting the chances of cardiac remodeling affecting the ECG or TTE.

Limitations

While Seamens’ Sign is a quick, effective, reliable alternative to other criteria for diagnosing LVH on ECG, there are study limitations. All ECGs were included, without removal of bundle branch blocks or other abnormal findings that could alter the results. However, this limitation was applied across all criteria in the study, helping to eliminate any differences in their application. Also, with the exception of the Cornell criteria which differentiates between sexes, we did not differentiate the application of the other criteria based on sex. This could hide differences in application of the criteria between sexes, but stays true to original application of these criteria. There were no changes in application of criteria based on age. There are known differences in ECG appearance based on age, including potential QRS amplitude changes (Levy et al., 1987). Since this was a retrospective study, there were multiple providers obtaining ECGs, multiple echocardiographers performing the TTEs, and multiple cardiologists reading the TTEs, which could lead to variability in ECG and TTE acquisition, interpretation, and reporting of LVH. The majority of TTE reports did not calculate a quantitative measurement of left ventricular mass which can contribute to variability during cardiologist interpretation of LVH, and possibly introduce bias. Finally, the correlations between the various ECG criteria were not calculated during this study. Given that the ECG criteria to evaluate LVH yield binary qualitative results (either yes or no for LVH), the comparison of the sensitivities and specificities of each criterion against one another is adequate since each criterion evaluating LVH was calculated for each individual patient and independently compared against each individual patients’ TTE results.

Conclusion

When compared to both the Sokolow-Lyon criteria and the Cornell criterion for men, Seamens’ Sign is noninferior in confirming LVH on ECG. Additionally, Seamens’ Sign has higher inter-rater agreement compared to both Sokolow-Lyon criteria as well as the Cornell criteria for men and women, possibly related to its ease of use.

Supplemental Information

Supplemental Information 1 Dataset

Click here for additional data file.

The authors would like to acknowledge Taylor Robinson, Lucas C. Wollenman, Tyler Pfister, Rand Pope, Aaron Azose, Olivia Henry, Jessica O’Shea, and Ansley Kunnath for their assistance in chart review and data acquisition.

Additional Information and Declarations

Competing Interests

Author Contributions

Human Ethics

Data Availability

The authors declare there are no competing interests.

Philip Walker conceived and designed the experiments, performed the experiments, analyzed the data, prepared figures and/or tables, authored or reviewed drafts of the article, and approved the final draft.

Cathy A. Jenkins conceived and designed the experiments, analyzed the data, prepared figures and/or tables, authored or reviewed drafts of the article, and approved the final draft.

Jeremy Hatcher performed the experiments, authored or reviewed drafts of the article, and approved the final draft.

Clifford Freeman performed the experiments, authored or reviewed drafts of the article, and approved the final draft.

Nickolas Srica performed the experiments, authored or reviewed drafts of the article, and approved the final draft.

Bryant Rosell performed the experiments, authored or reviewed drafts of the article, and approved the final draft.

Eriny Hanna performed the experiments, authored or reviewed drafts of the article, and approved the final draft.

Cooper March performed the experiments, authored or reviewed drafts of the article, and approved the final draft.

Charles Seamens conceived and designed the experiments, authored or reviewed drafts of the article, and approved the final draft.

Alan Storrow conceived and designed the experiments, analyzed the data, prepared figures and/or tables, authored or reviewed drafts of the article, and approved the final draft.

Nicole McCoin conceived and designed the experiments, prepared figures and/or tables, authored or reviewed drafts of the article, and approved the final draft.

The following information was supplied relating to ethical approvals (i.e., approving body and any reference numbers):

The Vanderbilt University Institutional Review Board approved this study (IRB#200150). This study meets 45 CFR 46.104 (d) category (4) for Exempt Review.

The following information was supplied regarding data availability:

The raw data is available in the Supplementary File.

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
