# Peer review of "Seamens’ Sign: a novel electrocardiogram prediction tool for left ventricular hypertrophy"

_PeerJ, doi:10.7717/peerj.13548_

## Round 0.1 · original submission · Major Revisions

Please address all concerns and criticisms thoroughly.

·

Basic reporting

Academic reporting of cases is generally reserved for bedside case presentations (usually not mentioned in the standard textbooks) do not catch enough attention to warrant publication. This article is definitely not one such.

Experimental design

no comment

Validity of the findings

Any possible impact of this novel clinical finding is hard to predict as it depebds upon post it might find in every-day work. However, the method of presentation is clear and straightforward enough to permit replication.

·

Basic reporting

No comment

Experimental design

No comment

Validity of the findings

No comment

Additional comments

Well prepared, clear and effectively structured article.

·

Basic reporting

1. Dr. Walker and colleagues developed a novel, quick, easy to use ECG screening criterion (Seamens’ Sign) for LVH. This new criterion was defined as the presence of QRS complexes touching or overlapping in two contiguous precordial leads. When compared to both the Sokolow-Lyon criteria and the Cornell criterion for men, Seamens’ Sign is noninferior in ruling in LVH on ECG. Additionally, Seamens’ Sign has higher inter-rater agreement compared to both Sokolow-Lyon criteria as well as the Cornell criteria for men and women.
2. The English writing is well.
3. The literature reviews are comprehensive.
4. Raw data were shared.

Experimental design

1. The study design was acceptable.
2. In the methods, the authors reported a high prevalence of echocardiographic LVH "35%", but did not mention the quantitative definition for echocardiographic LVH, possibly leading to a bias.
3. There were no baseline profiles of the patients, i.e. hypertension which might limit the applications.
4. I would like to see the correlations between various ECG criteria and Seamens sign for LVH. For instance, what are the consistency rates between Sokolow-Lyon (+) and Seamens sign (+) as well as Sokolow-Lyon (-) and Seamens sign (-)..... And what are the additional advantages of Seamens sign on the other ECG criteria for echocardiographic LVH?

Validity of the findings

1. The sensitivity and specificity using Sokolow-Lyon and Cornell ECG-based criteria were close to prior studies.
2. The PPV and NPV were low, limiting the applications.

Additional comments

In the current ECG machine, there are automatic interpretation software for LVH interpretation according to various ECG criteria, which could be helpful to the physicians and the utilization of Seamens signs might be limited.

·

Basic reporting

At first, this appeared as a decently written manuscript that would
find its way quite easily. ¸Unfortunately, I distinctly remembered that I have read this paper before.

Experimental design

-

Validity of the findings

-

Additional comments

-

---

## Round 0.2 · Minor Revisions

In the authors' response, they stated their paper strengths and limitations below, please adding these statements in the revised manuscript.

The correlations between the various ECG criteria were not calculated during this study, as it was planned for a future study of the Seamens’ Sign. With regards to this study though, given that the ECG criteria to evaluate LVH yield qualitative results, we believe that the comparison of the sensitivities and specificities is adequate since the various test methods are calculated for each patient against the TTE for each individual patient, which was defined as the gold standard in our manuscript.

·

Basic reporting

When I reviewed the original of this paper. I detected a huge amount of previously published materials published elsewhere by the same group of authors. It was not clearly said that it was not a peer-reviwed material, which was ok to submit. So, my criticism is chiefly to the journal. The authors' work seems fit for the publication.

Experimental design

The authors' work seems fit for the publication.

Validity of the findings

The authors' work seems fit for the publication.

Additional comments

The authors' work seems fit for the publication.

·

Basic reporting

As prior comments

Experimental design

As prior comments

Validity of the findings

As prior comments.

Additional comments

In the authors' response, they stated their paper strengths and limitations below, please adding these statements in the revised manuscript.

The correlations between the various ECG criteria were not calculated during this study, as it was planned for a future study of the Seamens’ Sign. With regards to this study though, given that the ECG criteria to evaluate LVH yield qualitative results, we believe that the comparison of the sensitivities and specificities is adequate since the various test methods are calculated for each patient against the TTE for each individual patient, which was defined as the gold standard in our manuscript.

·

Basic reporting

I feel this submission is acceptable

Experimental design

I feel this submission is acceptable

Validity of the findings

I feel this submission is acceptable

Additional comments

I feel this submission is acceptable

---

## Round 0.3 · accepted · Accept

Thank you for the opportunity to review your excellent work!

·

Basic reporting

No comment

Experimental design

No comment

Validity of the findings

No comment

·

Basic reporting

The paper is acceptable at current version.

Experimental design

The paper is acceptable at current version.

Validity of the findings

The paper is acceptable at current version.

Additional comments

The paper is acceptable at current version.

·

Basic reporting

I had no suggestions previously.

Experimental design

I had no suggestions previously.

Validity of the findings

I had no suggestions previously.

Additional comments

I had no suggestions previously.